# Adherence to a Mediterranean Diet and Bone Mineral Density in Spanish Premenopausal Women

**DOI:** 10.3390/nu11030555

**Published:** 2019-03-05

**Authors:** Jesús Pérez-Rey, Raúl Roncero-Martín, Sergio Rico-Martín, Purificación Rey-Sánchez, Juan D. Pedrera-Zamorano, María Pedrera-Canal, Fidel López-Espuela, Jesús M. Lavado-García

**Affiliations:** Metabolic Bone Disease Research Group, University of Extremadura, 10003 Cáceres, Spain; jperezrey@hotmail.com (J.P.-R.); rronmar@unex.es (R.R.-M.); prey@unex.es (P.R.-S.); jpedrera@unex.es (J.D.P.-Z.); mariapedreracanal@gmail.com (M.P.-C.); fidellopez@unex.es (F.L.-E.); jmlavado@unex.es (J.M.L.-G.)

**Keywords:** Mediterranean diet, adherence, bone mineral density, premenopausal women, quantitative bone ultrasound, dual-energy X-ray absorptiometry scan, peripheral quantitative computed tomography

## Abstract

The Mediterranean diet (MD) has been associated with an improvement in health and an increase in longevity. Certain components of a MD can play a role in the prevention of osteoporosis and/or hip fracture. We investigated the association between the degree of adherence to a MD and bone mineral density (BMD) measured in several bone areas in a population of Spanish premenopausal women. We analyzed 442 premenopausal women aged 42.73 ± 6.67 years. Bone measurements were obtained using quantitative bone ultrasound (QUS) for the phalanx, dual energy X-ray absorptiometry (DXA) for the lumbar spine, Ward’s triangle, trochanter, and hip, and peripheral quantitative computed tomography (pQCT) for the non-dominant distal forearm. MD adherence was evaluated with MedDietScore. Amplitude-dependent speed of sound (Ad-SOS), BMD, and volumetric bone mineral density (vBMD) (total, trabecular, and cortical bone density) were positively associated with higher adherence to the MD (*p* < 0.05). Adherence to the MD was significantly associated with QUS, BMD, and vBMD in multiple regression analysis; QUS: Ad-SOS (m/s) β = 0.099 (*p* = 0.030); BMD (g/cm^2^): femur neck β = 0.114 (*p* = 0.010) and Ward’s triangle β = 0.125 (*p* = 0.006); vBMD (mg/cm^3^): total density β = 0.119 (*p* = 0.036), trabecular density β = 0.120 (*p* = 0.035), and cortical density β = 0.122 (*p* = 0.032). We conclude that the adherence to the MD was positively associated with better bone mass in Spanish premenopausal women.

## 1. Introduction

Diet is a modifiable factor in the development and maintenance of bone mass [1]. Diet is considered to have an important role during childhood and adolescence, but there is no clear evidence on the effect of diet on bone mass in adults and in the elderly. Traditionally, research on bone metabolism has focused on the beneficial effects of calcium and vitamin D [2,3], as well as the controversial role of proteins in bone metabolism [4]. However, the beneficial effects of diet on bone health go beyond these nutrients [5].

A traditional Mediterranean diet (MD) is characterized by a high intake of vegetables, legumes, fruits, nuts, cereals, and unsaturated fats, especially olive oil; a low intake of saturated fats, meat, and poultry; a moderate to high intake of fish; a low to moderate intake of dairy products, usually in the form of cheese or yogurt; and a modest intake of alcohol in the form of wine [6,7,8]. There is consistent evidence that adherence to a traditional MD is associated with a reduction in the risk of all causes of mortality as well as with a reduction in the incidence of and mortality from coronary heart disease and cancer [9,10,11].

Osteoporosis is an important worldwide public health concern that affects millions of people [12,13]. This disease is characterized by low bone mass and the microarchitectural deterioration of bone tissue, with a consequent increase in bone fragility and susceptibility to fracture [14]. Although the mechanisms involved in the pathogenesis of osteoporosis are not fully understood yet, the genetic and environmental factors affecting osteoporosis development are currently being identified and are better known. Osteoporosis is mainly an age-related problem, with a higher prevalence among postmenopausal women [15], but bone modelling and remodelling are also influenced, among other factors, by genetics, nutrition, mechanics, hormones [16], and a sedentary lifestyle [17,18].

Several authors have indicated that the low incidence of osteoporosis in Mediterranean countries could be explained by diet [19,20,21]. The possible beneficial effects of some of the components of a Mediterranean diet, such as fruits, vegetables, and fish, on bone mass have been previously studied [5,22]; however, the association between MD and a risk of fractures is not yet clear [23,24].

In this cross-sectional study, we investigated the association between the degree of adherence to a MD and bone mineral density (BMD) measured in several bone areas by quantitative bone ultrasound (QUS), dual energy X-ray absorptiometry (DXA), and peripheral quantitative computed tomography (pQCT) in a population of Spanish premenopausal women.

## 2. Material and Methods

### 2.1. Study Population

This cross-sectional study was performed between 2014 and 2017. The participants were recruited from a clinical convenience sample from the area of Cáceres (Spain) and nearby communities via web advertising and primary care consults. All participants provided written informed consent, and the University of Extremadura Ethical Committee approved the research in accordance with the 1975 Declaration of Helsinki (approval code: 55/2015).

We aimed to have enough statistical power to detect medium-low effect sizes (anticipated Cohen’s ʄ = 0.15) with a β = 0.80 and α = 0.05, which required a minimum sample size of 432 participants [25]. A total of 442 healthy premenopausal women were included in this study (mean age: 42.73 ± 6.67 years). All the women were Caucasian, and none had dietary restrictions, neurological impairments, or physical disabilities; their medical histories showed no presence of low-trauma fractures. None of the women were using anti-osteoporotic drugs.

Before each candidate was enrolled in the study, a complete medical history was taken. Height was measured using a Harpenden stadiometer, and weight was measured using a biomedical precision balance. Both measurements were determined when participants were wearing only light clothing and no shoes. Body mass index (BMI) was calculated as the weight in kilograms divided by the square of the height in meters (kg/m^2^). We assessed the participants’ physical activity status on the basis of their answer to the following question [26]: “How much do you exercise or strain yourself physically in your leisure time?” The response categories were (1) sedentary (reading, watching television), (2) moderate (walking, cycling, and exercising in other ways for at least 4 h per week), (3) active (fitness-improving sport at least three times per week), and (4) competitive sport. Alcohol intake was sporadic, not exceeding 100 mL/day in any case. In total, 68.6% of the participants were non-smokers (*n* = 303).

### 2.2. Dietary Assessment and Mediterranean Diet Adherence

Nutrient intake was quantified using a 131-item food frequency questionnaire (FFQ) that has been previously validated [15]. This FFQ used dietetic scales, measuring cups, cans, small bottles, and spoons, on the basis of current 7-day dietary records, as in previous studies [5,15]. The questionnaire used was self-reported, and the person completing the interview was blinded to the research question and hypothesis. Using this FFQ, we assessed the dietary intake of carbohydrates, protein, fat, calcium, and vitamin D from the Spanish Food Composition database [27].

MD adherence was assessed using the questionnaire proposed by Panagiotakos et al. in 2006 [28]. The questionnaire consisted of 11 items (non-refined cereals, fruits, vegetables, legumes, potatoes, fish, meat and meat-based products, poultry, full-fat dairy products, olive oil, and alcohol), which were scored from 0–5 based on the frequency of consumption and on the level of adherence to the Mediterranean dietary pattern. In particular, for the consumption of non-refined cereals, fruits, vegetables, legumes, olive oil, fish, and potatoes, a score of 0 is assigned when someone reports no consumption, a score of 1 is assigned when the consumption of 1–4 servings/month is reported, a score of 2 for 5–8 servings/month, a score of 3 for 9–12 servings/month, a score of 4 for 13–18 servings/month, and a score of 5 for more than 18 servings/month. On the other hand, for meat and meat products, poultry, and full-fat dairy products, the scores are assigned on a reverse scale. For alcohol, a monotonic function is not used; instead, a score of 5 is assigned for the consumption of less than 300 mL of alcohol/day, a score of 0 is assigned for the consumption of more than 700 mL/day, and scores 4–1 are assigned for the consumption of 300, 400–500, 600, and 700 mL/day (100 mL has 12 g ethanol concentration), respectively. Thus, the total score ranged from 0 to 55, with higher scores showing greater adherence to the MD. The participants were stratified according to tertiles of MD adherence scores: Low (<29), Medium (>29 and <32), and High (>32).

### 2.3. Measurement of Bone Mineral Density

Ultrasounds were performed on the second to the fifth proximal phalanx of the non-dominant hand using a DBM Sonic Bone Profiler (IGEA, Capri, Italy), which measures amplitude-dependent speed of sound (Ad-SOS). The femoral neck and L2–L4 vertebrae BMDs were analyzed via DXA (Norland XR-800, Norland Inc., Fort Atkinson, WI, USA), and the measurements were expressed as the quantity of mineral divided by the area scanned (g/cm^2^). pQCT measurements were performed on the non-dominant distal forearm using a Stratec XCT-2000 device (Stratec Medizintechnik, Pforzheim, Germany).

### 2.4. Statistical Analysis

The continuous variables are presented as the means ± standard deviation (SD), and the categorical variables are presented as frequencies and percentages. Some of the studied variables were not normally distributed; thus, when appropriate, a two-step approach was used to normalize the data before statistical analyses were conducted [29]. This approach included transforming the variable into a percentile rank, and after, we examined the transformed versions of those variables. The participants were categorized into three groups according to tertiles of MD adherence scores: Low (<29), Medium (>29 and <32), and High (>32). The Pearson chi-square test was used to compare categorical variables between groups, and an analysis of variance (ANOVA) was used to compare continuous variables between groups. In addition, the mean BMD values were compared between groups using an analysis of covariance (ANCOVA), controlling for age (years), menarche age (years), BMI (kg/m^2^), energy (kcal/day), calcium (mg/day), vitamin D (µg/day), physical activity (sedentary, moderate, and active), and smoking. A multiple linear regression (using the stepwise method) was used to examine whether the studied variables, age (years), menarche age (years), BMI (kg/m^2^), calcium intake (mg/day), vitamin D intake (µg/day), energy intake (kcal/day), physical activity, smoking, and Mediterranean diet score, were predictive of selected BMD parameters. All of the analyses were performed with SPSS 24.0 for Windows (SPSS, Inc., Chicago, IL, USA).

## 3. Results

A total of 442 women participated in the study. The main characteristics of the participants in terms of biological status, anthropometric values, and dietary intake are shown in Table 1. Based on the tertiles of the MD adherence scores, there were significant differences between groups in smoking status (*p* = 0.021) and Ca intake (*p* = 0.025). There were no significant differences in the other analyzed variables.

Table 2 describes the associations between the tertiles of adherence to MD and QUS, BMD, and volumetric BMD. We found statistically significant differences in Ad-SOS (*p* = 0.008), femoral neck (*p* = 0.004), Ward’s triangle (*p* = 0.001), lumbar spine (*p* = 0.029), and volumetric BMD total density (*p* = 0.006) and cortical density (*p* = 0.007). Post hoc tests indicated that there were significant differences between tertile 1 and tertile 2 in femoral neck BMD, Ward’s triangle BMD, and lumbar spine BMD and between tertile 1 and tertile 3 in Ad-SOS, femoral neck BMD, Ward’s triangle BMD, and volumetric BMD total density and cortical density. After adjustments for age (years), menarche age (years), BMI (kg/m^2^), energy (kcal/day), calcium (mg/day) vitamin D (µg/day), physical activity (sedentary, moderate, and active), and smoking, there were statistically significant differences in all bone density variables studied (*p* < 0.05). Post hoc tests indicated that there were significant differences between tertile 1 and tertile 2 in Ad-SOS, femoral neck BMD, trochanter BMD, and Ward’s triangle BMD and between tertile 1 and tertile 3 in Ad-SOS, femoral neck BMD, Ward’s triangle BMD, and volumetric BMD total density, trabecular density, and cortical density.

We further explored BMD density by multiple regression analysis (Table 3). The MD score was positively associated with Ad-SOS (β = 0.099; *p* = 0.030), femoral neck BMD (β = 0.114; *p* = 0.010), Ward’s triangle BMD (β = 0.125; *p* = 0.006), volumetric BMD total density (β = 0.119; *p* = 0.036), volumetric BMD trabecular density (β = 0.120; *p* = 0.035), and volumetric BMD cortical density (β = 0.122; *p* = 0.032) but was not associated with trochanter BMD (β = 0.081; *p* = 0.073) and L2–L4 BMD (β = 0.047; *p* = 0.326).

## 4. Discussion

In the present study, we found a positive association between the degree of adherence to MD and bone density assessed by QUS, pQCT, and DXA in a population of Spanish premenopausal women. Although DXA is the gold standard technique for the diagnosis of osteoporosis, the use of pQCT in radius assessment provides a measure of volumetric bone mineral density (vBMD) and distinguishes trabecular from cortical bone. QUS is an alternative and/or integrative technique to DXA scan; it is a radiation-free, transportable technique that uses sound waves to evaluate bone properties that are not measured by the DXA scan.

While a MD has been associated with a lower risk for many chronic diseases in populations worldwide, its associations with bone health and especially BMD are less well-known; however, these relationships have been analyzed in recent years [30]. Most studies have been conducted on the elderly, especially elderly women. Our study was carried out with premenopausal women, which is a population group of which there are few data regarding the relationship between a MD and bone health.

The favorable associations observed in our study were consistent with those in several other studies but not those in all studies. In our study, increased adherence to a MD was associated with higher BMD values measured by phalanx QUS, radius pQCT, and DXA in the femoral neck, Ward’s triangle, and lumbar spine. Similar findings have been reported in European studies [21,31,32], but bone density was determined by calcaneus DXA and calcaneus QUS. In addition, in a Chinese population, bone density was evaluated by DXA in the femoral neck, lumbar spine, trochanter, and Ward’s triangle, and it was found that greater adherence to a MD was associated with higher BMD in the middle-aged and elderly populations [33]. In an American population, this association was also found [34,35]. However, Kontogianni et al. [20] found no association between MD adherence and bone parameters determined with DXA in 220 pre- and postmenopausal women; however, they observed an association between these parameters and dietary patterns characterized by high intakes of fish and olive oil, as well as a low consumption of red meat.

The potential protective effect of MD adherence has been positively related to the risk for hip fractures [23,34,36,37,38]. However, other authors did not observe this relationship [24,39].

Our results are consistent with the findings of a meta-analysis published in 2018 by Malmir et al. [30] that included 13 studies carried out in different European countries, China, and the USA. They evaluated the associations between MD adherence and BMD as well as the risk for fractures. The meta-analysis found that a high adherence to a MD was associated with a 21% reduction in the risk for hip fractures and was positively associated with the BMD of the spine and femoral neck. Currently, there are no studies that associate MD adherence with BMD measured by pQCT and phalanx ultrasound; therefore, our study is the first to relate MD adherence to these two techniques.

The results of our study could be attributed to the combination of the beneficial effects of the individual MD components. It is widely accepted that nutrition, as well as being a modifiable factor, is an essential factor in bone health, both to achieve and maintain optimal bone mass and to prevent osteoporosis [40,41]. Specific components of this diet may play a role in the prevention of osteoporosis and/or in the prevention of different types of fractures [42]. Thus, several studies have found that a moderate consumption of fish could be useful in maintaining an adequate bone mass [5] due to its contribution of high quality proteins, polyunsaturated fatty acids n-3 (PUFAs), vitamins A and D, and minerals such as selenium, calcium, iodine, and zinc [43]. Some of these nutrients have beneficial effects on bone health. The positive effect of calcium and vitamin D is recognized, and it is known that in Spain a high amount (87%) of vitamin D intake comes from the consumption of fish [27]. Zinc in the diet could also be positively correlated with BMD [44], and PUFAs are associated with an increase in bone mass [45,46,47].

The protective effect of a MD against various diseases seems to be mediated by the anti-inflammatory properties of some of its beneficial components, such as the polyphenols, present in foods such as vegetables, fruits, and red wine [48]. A study carried out in Spain and published in 2012 [49] found that a MD had a positive effect on serum markers of bone formation and a reduction in the circulating concentration of markers of bone resorption when the MD was enriched with virgin olive oil. A dose-dependent protective effect of oleuropein, which is a polyphenol from olive oil, has been found in an in vivo experimental model of rats with bone loss [19]. A recent study concluded that the dietary intake of olive oil is positively associated with total, trabecular, and cortical BMD [50].

Our work has several limitations. First, the design of the study was cross-sectional and therefore can only indicate associations and not causality. Second, all of the participants were recruited in our region and surrounding areas; thus, the different patterns of MD adherence across Spain might not be represented. Third, we measured physical activity with a questionnaire that might not efficiently quantify it. Fourth, some of the results in which statistical significance was not obtained but was expected could be reflecting a lack of statistical power (type 2 error). Finally, there are several methods to calculate and evaluate adherence to the MD [51], so the degree of adherence to the MD in our study population may not be similar to the results from the use of another tool. However, the questionnaire used has proven to be an effective tool [28] and has been shown to be comparable with other Med Scores in previous studies [51]. This study also has strengths that need to be highlighted. This research provided a comprehensive examination of the associations between several modifiable factors that can influence BMD in premenopausal women. In addition, the measurements that evaluated BMD with three important assessment techniques (DXA, pQCT, and ultrasound) and scans of different areas of the body allowed us to gain a better understanding of their association with MD; these techniques are widely validated, and therefore the accuracy of the results is guaranteed. This study is the first to relate MD adherence with BMD measured by pQCT and phalanx ultrasound.

In conclusion, we found that better MD adherence was positively associated with BMD measured by DXA, pQCT, and phalanx ultrasound in Spanish premenopausal women. We encourage further investigations, particularly longitudinal studies or clinical trials that consider the observed association.

## Figures and Tables

**Table 1 nutrients-11-00555-t001:** Biological, anthropometric, and dietetic factors in the study according to tertile of Mediterranean diet score.

	Tertile 1 (*n* = 138)	Tertile 2 (*n* = 122)	Tertile 3 (*n* = 182)	*p*-Value
Age (years)	43.10 ± 6.10	43.33 ± 6.64	42.04 ± 7.07	0.190
Menarche Age (years)	12.75 ± 1.39	12.75 ± 1.34	12.48 ± 1.34	0.129
Gravidity (*n*)	1.80 ± 1.35	1.71 ± 1.23	1.72 ± 1.08	0.773
Births (*n*)	1.61 ± 1.12	1.49 ± 0.94	1.53 ± 0.94	0.643
Breastfeeding (months)	5.33 ± 7.10	5.16 ± 5.65	5.50 ± 6.52	0.904
Weight (kg)	63.55 ± 9.36	63.08 ± 7.95	62.87 ± 8.62	0.782
Height (m)	1.59 ± 0.05	1.59 ± 0.06	1.59 ± 0.06	0.708
BMI (kg/m^2^)	24.94 ± 3.21	24.67 ± 2.76	24.77 ± 3.17	0.770
Smoking				
No	84 (60.9%)	82 (67.2%)	137 (75.3%)	0.021 ^b^
Yes	54 (39.1%)	40 (32.8%)	45 (24.7%)	
Physical Activity				
Sedentary	64 (46.4%)	42 (34.4%)	80 (44.0%)	0.161
Moderate	32 (23.2%)	26 (21.3%)	42 (23.1%)	
Active	42 (30.4%)	54 (44.3%)	60 (33.0%)	
Dietary Intake				
kcal/day	2290.50 ± 620.07	2301.34 ± 664.23	2182.60 ± 638.93	0.189
Proteins (g/day)	87.38 ± 27.55	93.71 ± 34.84	88.58 ± 30.10	0.214
Carbohydrates (g/day)	297.12 ± 109.07	288.83 ± 95.16	281.25 ± 99.86	0.386
Fats (g/day)	83.42 ± 24.00	84.86 ± 33.46	79.82 ± 31.07	0.310
Vitamin D (µg/day)	6.70 ± 5.31	7.95 ± 8.00	7.03 ± 5.38	0.243
Ca (mg/day)	1026.46 ± 520.07	1201.36 ± 579.74	1112.03 ± 463.32	0.025 ^a^

BMI: body mass index. ^a^ Tertile 1 vs. Tertile 2; ^b^ Tertile 1 vs. Tertile 3

**Table 2 nutrients-11-00555-t002:** Bone density by tertiles of Mediterranean diet score.

	Tertile 1 (*n* = 138)	Tertile 2 (*n* = 122)	Tertile 3 (*n* = 182)	*p*-Value	*p*-Value ^&^
Quantitative Bone Ultrasound					
Ad-SOS (m/s)	2112.23 ± 48.88	2126.25 ± 49.91	2128.58 ± 47.40	0.008 ^b^	0.001 ^a,b^
BMD (g/cm^2^)					
BMD Femur Neck	0.859 ± 0.126	0.902 ± 0.121	0.902 ± 0.123	0.004 ^a,b^	0.001 ^a,b^
BMD Trochanter	0.665 ± 0.112	0.688 ± 0.098	0.683 ± 0.107	0.172	0.036 ^a^
BMD Ward’s Triangle	0.650 ± 0.123	0.696 ± 0.129	0.701 ± 0.121	0.001 ^a,b^	0.001 ^a,b^
BMD Lumbar Spine (L2–L4)	1.044 ± 0.147	1.091 ± 0.140	1.070 ± 0.139	0.029 ^a^	0.009 ^a^
Volumetric BMD (mg/cm^3^)					
Total Density	346.45 ± 50.19	357.21 ± 50.62	368.46 ± 48.02	0.006 ^b^	0.001 ^b^
Trabecular Density	174.82 ± 36.25	182.85 ± 39.18	185.47 ± 33.50	0.095	0.013 ^b^
Cortical Density	485.23 ± 69.59	503.89 ± 71.53	516.77 ± 74.31	0.007 ^b^	0.003 ^b^

BMD: body mass density. ^a^ Tertile 1 vs. Tertile 2; ^b^ Tertile 1 vs. Tertile ^3^. ^&^ After further adjustment for age (years), menarche age (years), BMI (kg/m^2^), energy (kcal/day), calcium (mg/day), vitamin D (µg/day), physical activity (sedentary, moderate, and active), and smoking.

**Table 3 nutrients-11-00555-t003:** Multiple linear regression analysis for the association between bone density and age (years), menarche age (years), BMI (kg/m^2^), calcium intake (mg/day), vitamin D intake (µg/day), energy intake (kcal/day), physical activity, smoking, and Mediterranean diet score.

Quantitative Bone Ultrasound: Ad-SOS (m/s)			
Optimal Model	*R* ^2^	Adjusted *R*^2^	
	0.115	0.111	
Selected Independent Variable	Standardized β	*t*	*p*-value
BMI (kg/m^2^)	−0.326	−7.188	<0.001
Mediterranean Diet Score	0.099	2.171	0.030
**BMD Femur Neck (g/cm^2^)**			
Optimal Model	*R* ^2^	Adjusted *R*^2^	
	0.174	0.168	
Selected Independent Variable	Standardized β	*t*	*p*-value
BMI (kg/m^2^)	0.391	8.703	<0.001
Age (years)	−0.183	−4.061	<0.001
Mediterranean Diet Score	0.114	2.584	0.010
**BMD Trochanter (g/cm^2^)**			
Optimal Model	*R* ^2^	Adjusted *R*^2^	
	0.147	0.143	
Selected Independent Variable	Standardized β	*t*	*p*-value
BMI (kg/m^2^)	0.392	8.597	<0.001
Age (years)	−0.108	−2.368	0.018
Mediterranean Diet Score	0.081	1.794	0.073
**BMD Ward’s Triangle (g/cm^2^)**			
Optimal Model	*R* ^2^	Adjusted *R*^2^	
	0.140	0.134	
Selected Independent Variable	Standardized β	*t*	*p*-value
Age (years)	−0.302	−6.568	<0.001
BMI (kg/m^2^)	0.241	5.264	<0.001
Mediterranean Diet Score	0.125	5.264	0.006
**BMD Lumbar Spine: L2–L4 (g/cm^2^)**			
Optimal Model	*R* ^2^	Adjusted *R*^2^	
	0.066	0.064	
Selected Independent Variable	Standardized β	*t*	*p*-value
BMI (kg/m^2^)	0.256	5.499	<0.001
Mediterranean Diet Score	0.047	0.983	0.326
**Volumetric BMD: Total Density (mg/cm^3^)**			
Optimal Model	*R* ^2^	Adjusted *R*^2^	
	0.050	0.040	
Selected Independent Variable	Standardized β	*t*	*p*-value
Menarche Age (years)	−0.140	−2.476	0.014
Mediterranean Diet Score	0.119	2.108	0.036
BMI (kg/m^2^)	0.14	2.010	0.045
**Volumetric BMD: Trabecular Density (mg/cm^3^)**			
Optimal Model	*R* ^2^	Adjusted *R*^2^	
	0.026	0.020	
Selected Independent Variable	Standardized β	*t*	*p*-value
Mediterranean Diet Score	0.120	2.114	0.035
BMI (kg/m^2^)	0.114	2.003	0.046
**Volumetric BMD: Cortical Density (mg/cm^3^)**			
Optimal Model	*R* ^2^	Adjusted *R*^2^	
	0.042	0.035	
Selected Independent Variable	Standardized β	*t*	*p*-value
Menarche Age (years)	−0.158	−2.794	0.006
Mediterranean Diet Score	0.122	2.155	0.032

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
