# Peer review of "Adherence to a Mediterranean Diet and Bone Mineral Density in Spanish Premenopausal Women"

_nutrients, 2019, doi:10.3390/nu11030555_

Reviewer 1 Report

The research topic on Mediterranean diet and bone health including osteoporosis and fracture risk has been addressed in different populations. There have been individual cohort studies as well as systematic review and pool-analyses published in the literature. Hence, the current study did not seem to add much information to address this topic. 

Regarding methodology, a 7-day dietary record is a weakness. There is not much detail how the FFQ was derived from this one week dietary record and what were the food items included in the FFQ. Has the FFQ been validated in another population to ensure it reproducibility? The authors also need to provide more details about how the Med score was determined using the citation the mentioned. And how was this score comparable to other Med scores in the literature? 

The models used for statistical analysis did not seem to be appropriate. The authors presented R square for model fit but the results in the abstract presented beta coefficient for the linear relationship between the Med score and various bone parameters. 

For the results, it seems strange that those who consumed a Med diet had lower dietary vitamin D and Ca. I thought the relationships with these two nutrients would be positive, since those who adhered the Med diet would have a sufficient or at least higher intakes for these two nutrients. 

Author Response

Point 1: The research topic on Mediterranean diet and bone health including osteoporosis and fracture risk has been addressed in different populations. There have been individual cohort studies as well as systematic review and pool-analyses published in the literature. Hence, the current study did not seem to add much information to address this topic.

Response: Our study is interesting in two aspects:

1) Study population in which it has been carried out are premenopausal women. There are few studies on this population , so the article adds some information.

2) This study is the first to relate MD adherence with BMD measured by pQCT and phalanx ultrasound (QUS). Although DXA is the gold standard technique for the diagnosis of osteoporosis, pQCT in radius assessment provides a measure of volumetric bone mineral density (vBMD) and distinguishes trabecular from cortical bone. QUS is an alternative and/or integrative technique to DXA scan; it is a radiation-free, transportable technique that uses sound waves to evaluate bone properties that are not measured by the DXA scan.

Point 2: Regarding methodology, a 7-day dietary record is a weakness. There is not much detail how the FFQ was derived from this one week dietary record and what were the food items included in the FFQ. Has the FFQ been validated in another population to ensure it reproducibility?

Response: Following the reviewers suggestions, we have added this recommendation to the methodology section.  This FFQ record: olive oil and other cooking oils, drinks (coffee, coke, juices…), meat and meat-based products, sweets, fruits, milk and milk-derived products, fish, legumes and vegetables, alcoholic drinks, breads, pasta, etc. This FFQ has been used in another population (posmenopausal women, men and kids; healthy and unhealhy.

The authors also need to provide more details about how the Med score was determined using the citation the mentioned. And how was this score comparable to other Med scores in the literature?

Response:  Following the reviewers suggestions, we have included this suggestion in the methodology section. Med Score has proven to be an effective tool and comparable with other Med scores in previous studies [Public Health Nutr. 2011 Dec;14(12A):2338-45 and Nutr. Metab. Cardiovasc. Dis. 2006, 16, 559–568]

Point 3: The models used for statistical analysis did not seem to be appropriate. The authors presented R square for model fit but the results in the abstract presented beta coefficient for the linear relationship between the Med score and various bone parameters. 

Response: the results presented in the abstract are the same as those in the table 3. We highlight the influence of the Med Score within the multiple regression model in the BMD, reason why we present the beta coefficient.

Point 4: For the results, it seems strange that those who consumed a Med diet had lower dietary vitamin D and Ca. I thought the relationships with these two nutrients would be positive, since those who adhered the Med diet would have a sufficient or at least higher intakes for these two nutrients.

Response: These results can be explained by the fact that there is no direct relationship between some foods rich in vitamin D or calcium and the maximum score on the item valued. Example: intake of milk with a score of 5 when it was never consumed. Regarding vitamin D, fish consumption in Spain provides 87% of the total daily intake of vitamin D (Eur J Clin Nutri. 2010; 64, S37-S43). In our study, not all adherents to the Mediterranean diet should have to eat  fish.

Reviewer 2 Report

This was a cross sectional study to examine the independent influence of Mediterranean diet (MD) adherence on bone health in pre-menopausal women. The data suggested that dietary factors apart from calcium and vitamin D may have important influences on bone health in pre-menopausal women. There are suggestions to improve on the paper:

1.     The percentage of smokers in this group of women was relatively high (about one third). This should be mentioned in discussion as MD may have protected bone from the negative effects of smoking. It is puzzling why the variables considered in the stepwise linear regression were different from those in Table 2, and why smoking which was significantly associated with MD adherence was not included in the model.

2.     It was said to be a strength of the study to include different methods of measurement at different sites. But how they each contribute to the understanding of overall bone health and why some bone measurements e.g. lumbar spine BMD did not show independent influence of MD were not discussed.

3.     Physical activity is an important confounder factor. But the measurement appeared to be quite crude. This should be stated as a limitation.

4.     In method, it is not clear whether the food frequency questionnaire was based on a one off interview by a trained research assistant, or a self-administered questionnaire or 7 day food record. It was said to have been used to estimate calcium and vitamin D intake. But it appeared to have been used to estimate macronutrients e.g. carbohydrate and protein as well.

Author Response

Point 1: The percentage of smokers in this group of women was relatively high (about one third). This should be mentioned in discussion as MD may have protected bone from the negative effects of smoking.

Response: We did not include the percentage in discussion because in the stepwise linear regression, smoking had not influenced BMD.   

It is puzzling why the variables considered in the stepwise linear regression were different from those in Table 2, and why smoking which was significantly associated with MD adherence was not included in the model.

Response:  variables considered in the stepwise linear regression were similar to the ones  in Table 2. By mistake, we omitted smoking and physical activity in the text. Now, we include it in the text.

Point 2:  It was said to be a strength of the study to include different methods of measurement at different sites. But how they each contribute to the understanding of overall bone health and why some bone measurements e.g. lumbar spine BMD did not show independent influence of MD were not discussed.

Response: We agree with the reviewer on the fact that if there is an association with MD in all body areas, there should also be in lumbar spine. However, due to the anatomical and morphological characteristics of this area it could not be detected as a consequence of  a lack of statistical power because the effect of adherence to MD is lower in this area.

Following the reviewer suggestion, we include it in limitations.

Point 3:   Physical activity is an important confounder factor. But the measurement appeared to be quite crude. This should be stated as a limitation.

Response: Following the reviewer suggestion, we include it in limitation.

Point 4:   In method, it is not clear whether the food frequency questionnaire was based on a one off interview by a trained research assistant, or a self-administered questionnaire or 7 day food record.

Response:  

Following the reviewer suggestion, we rewrote this part of the manuscript.

It was said to have been used to estimate calcium and vitamin D intake. But it appeared to have been used to estimate macronutrients e.g. carbohydrate and protein as well.

Response:  FFQ was used to estimate dietary intake of carbohydrates, protein, fat, calcium and vitamin D. We include it in the manuscript. While not perfect, FFQs have been validated to estimate energy, macronutrients and micronutrients [Am. J. Epidemiol. 1985, 122, 51–65; Ann. Epidemiol. 1999, 9, 178–187;]

Round  2

Reviewer 1 Report

Many questions I raised earlier have not been addressed sufficiently to improve the manuscript.. 

Author Response

Following the reviewers suggestions, we have improved some manuscript sections